# CRAFT: CROSS-REPRESENTATION MODELING ON AUDIO WAVEFORMS AND SPECTROGRAMS

## ABSTRACT

In this paper, we introduce Cross-Representation modeling on Audio waveForms and specTrograms (CRAFT), an innovative representation modeling designed to extract joint features from diverse representations in the audio modality, and choose acoustic classification to showcase the effectiveness of our approach. Historically, most prior works are focused on utilizing either the frequency-domain spectrogram or the time-domain waveform representations for acoustic modeling. Directly fusing or concatenating individual representations suffers from performance degradation. However, we argue that by aligning these individual representations effectively, they can complement each other and substantially enhance the quality of downstream tasks. To mitigate semantic misalignment, we initially propose a cross-representation contrastive learning framework incorporating spectrogram and waveform based contrastive learning loss in audio pre-training. Subsequently, to alleviate temporal misalignment, we present a cross-representation transformer architecture, which models on spectrogram and waveform tokens together with fusion bottlenecks. The proposed CRAFT is tested on two commonly used datasets, demonstrating superior performances. Notably, our proposed CRAFT method outperforms the spectrogram-based counterpart by an impressive 4.4% higher mAP on AudioSet balanced set, and achieves SOTA comparable performances on full set, which suggests the alleviation of semantic misalignment and temporal misalignment boosts cross-representation performances in audio modeling. All codes and models will be open-sourced.

## 1 INTRODUCTION

In previous studies, it has been a common practice to treat distinct domain representations within the same modality as if they were entirely separated modalities. To illustrate this, consider the audio domain: spectrogram-based frequency representations are frequently employed for audio classification tasks (Gong et al., 2021), while time-domain representations are commonly utilized for tasks such as audio signal processing (Purwins et al., 2019) and speech separation (Luo & Mesgarani, 2018). Currently, the majority of the audio agging approaches, including Recurrent Neural Network (RNN) (Freitag et al., 2017), Convolutional Neural Network (CNN) (Zhu et al., 2018), Transformer (Gong et al., 2021), take spectrogram-based features as inputs, such as log-mel features and MFCC features. However, energy map based approaches potentially lead to information loss. As a comparison, raw waveforms are proved to convey complement information as opposed to audio analytical features (Yao et al., 2018; Sainath et al., 2015). The aggregation of spectrograms and waveforms in the era of transformer has the potential towards performance enhancement but still remains unclear.

Fusing information from diverse domain representations, such as waveform and spectrogram representations in the context of audio, poses non-trivial challenges. These challenges primarily revolve around two critical aspects: (i) Semantic Misalignment: The features acquired from spectrograms predominantly emphasize time-frequency responses, whereas waveform-based features concentrate more on capturing common time-amplitude patterns. For instance, it might be easier to locate the explosive car engine sound by detecting frequency changes, and successive breezes might be more accurately reflected on amplitude changes. Consequently, employing a straightforward approach of naively fusing or concatenating features from distinct modalities fails to appreciably enhance the comprehensiveness of the feature set, thereby hindering improvements in classification performances. (ii) Temporal Misalignment: Spectrograms are generated using Short-Time Fourier Trans-

form (STFT), which imposes fixed frequency and temporal resolutions. In contrast, waveform representations involve direct learning from raw samples and aggregation at multiple scales. Thus, instead of simply concatenating features from different representations over time, we must devise a methodology to enhance the representation alignment, accounting for these dissimilar temporal characteristics.

To tackle the aforementioned challenges, we introduce CRAFT, a tailored two-step system meticulously designed to improve both semantic and temporal alignment among representations originating from distinct calculations within the audio modality. In our pursuit of semantic alignment, we introduce an innovative cross-representation contrastive learning approach (named PSWaCL in Section. 3.3) during the pretraining phase, in addition to the conventional single representation learning tasks. In addition to the original audio modeling on spectrograms, PSWaCL further contrasts spectrogram and waveform representations in pretraining, and learns diverse representations from them simultaneously. This method plays a pivotal role in enhancing the alignment of feature semantics derived from waveform and spectrogram representations. To achieve cross-representation temporal alignment, we draw inspiration from the concept of bottleneck tokens to facilitate the fusion of features across domains (named SWaB in Section. 3.4) in downstream tasks. To tackle the misalignment of spectrogram tokens and waveform tokens, SWaB utilized fusion bottleneck to exchange features between them on each transformer layer. Our investigation into cross-representation strategies is conducted with tailored design, culminating in the presentation of strong performances for the application of contrastive loss in pretraining stage and bottleneck fusion in finetuning stage.

We conducted experiments on two frequently utilized datasets, achieving state-of-the-art (SOTA) or SOTA comparable performances. It is noteworthy that our innovative PSWaCL objective in pretraining and SWaB fusion in finetuning allow us to leverage waveform representations, leading to a remarkable 4.4% mAP increase in AudioSet balanced set (Gemmeke et al., 2017) classification, compared with the spectrogram-only counterpart. Furthermore, our proposed method exhibits robust generalization capabilities, delivering performance levels comparable to the state-of-the-art on downstream tasks, as evidenced by results on the ESC-50 dataset. Our contributions are:

- We propose SWaCL loss, which treats spectrogram and waveform representations as data augmentations to each other, and contrasts them in self-supervised audio modeling.

- Incorporated SWaCL loss, we further propose Cross-Representation modeling on Audio waveForms and specTrograms (CRAFT), which is equipped with (i) a multi-scale embedding on waveform inputs (named MSAE), (ii) a contrastive learning on spectrogram and waveform representations in pretraining (named PSWaCL), and (iii) bottleneck fusion in finetuning (named SWaB). To the best of our knowledge, CRAFT is the first transformer framework that systematically explores the cross-spectrogram-waveform modeling.

- We further conduct numerous experiments, empirically verify the design choices of CRAFT, and ablate the performance impact of each factor. With additional audio modeling on waveforms, CRAFT successfully achieves superior performances compared to spectrogram-only counterpart, and further proves that spectrograms and waveforms convey complement information. Our pioneering work on cross-representation modeling serves as reference for future applications in other modalities.

## 2  RELATED WORK

**Audio classification methods and cross-representation learning in Audio.** In the past decade, we have seen emerging approaches in classifying audio inputs, among which CNNs and transformers are the most popular methods. The success of transfer learning from vision domain to audio domain for CNN models was explored in (Grzywczak & Gwardys, 2014; Palanisamy et al., 2020). Since 2021, AST (Gong et al., 2021), HTS-AT (Chen et al., 2022) started to exploit convolution-free, purely attention-based methods in classifying audio spectrogram, and have shown superior performances over CNNs or CNN-attention hybrid models.

It has been demonstrated in many works that combining various representations, such as spectrogram and waveform, can improve the audio modeling results over single representation modeling. In the era of CNNs, works such as (Li et al., 2019; Fedorishin et al., 2021; Su et al., 2019; Yang & Hirschberg, 2018; Tokozume & Harada, 2017; Sainath et al., 2015) have proposed several multi-

stream models that simultaneously take waveform, spectrogram or even "motion features" (Yin et al., 2018) as inputs. Superior performances have been achieved from these methods over waveform-only or spectrogram-only counterparts. However, to the best of our knowledge, contrasting multi-representations instead of simply combining them, still remains unexplored, which is an unique contribution in this paper.

**Self-supervised learning.** The absence of labeled data in real-world datasets motivates the development of self-supervised learning (SSL) approaches, a research area extensively explored in the domains of Computer Vision (CV) and Natural Language Processing (NLP). One prominent avenue in SSL is Contrastive Learning (CL), which augments the input sample with different methods, and naturally pairs the resulting augmentations. Among numerous works, MoCo works (He et al., 2020; Chen et al., 2020c; 2021) and SimCLR (Chen et al., 2020a;b) works are noteworthy methods. Pioneering algorithms, such as MoCo and SimCLR, have significantly enhanced the performances of self-supervised pretraining, thus rendering SSL a good fit for large scale applications.

SimCLR (Chen et al., 2020a) represents a prominent exemplar of negative example based contrastive learning methods, where two streams share the same backbone. $N$ instances are sampled in each minibatch, where a contrastive prediction task is formulated on pairs of augmented instances, yielding a total of $2N$ instances. While SimCLR does not explicitly select negative instances, the remaining $2(N-1)$ augmented instances in the mini-batch are treated as negatives. In CRAFT, we assume that spectrogram and waveform contain common as well as complement information, and they form a natural pair of two augmented views given one input audio sample. Thus the design of SimCLR aligns well for contrasting audio representations and enables better instance discrimination in audio SSL, which benefits finetuning in downstream tasks.

**Multimodal task.** Multimodal task forms a challenging problem and a facilitator of human perceptual learning. Due to various reasons such as modal-specific input representations and the misalignment between different modalities, it is never a trivial problem to exchange shareable information towards building a unified model in resolving multimodal task (Jaegle et al., 2021; Gong et al., 2022b; Owens et al., 2016). Among various multimodal fusion works, MBT (Nagrani et al., 2021) delicately incorporated attention bottleneck tokens to fuse information between modalities, and restricts the flow of cross-modal information exchange. While MBT was originally proposed in the multimodal sceranio, we note that derived from the same audio sample, spectrograms and raw waveforms are mostly diversified audio inputs with specialised formats. Log-mel spectrograms are computed on every temporal interval, encompassing multiple frequency bins, while raw waveforms are essentially 1-dimensional vectors. Upon initial examination, they share minimal apparent commonalities, and are misaligned on temporal and frequency responses. Consequently, cross-representation within audio domain might leverage MBT during joint spectrogram-waveform training.

## 3 CRAFT: CROSS-REPRESENTATION MODELING ON AUDIO WAVEFORMS AND SPECTROGRAMS

Our work is built upon SSAST (Gong et al., 2022a), which operates on audio spectrograms, employs a masked spectrogram patch modeling (MSPM) framework and improves pretraining for various audio and speech tasks. In this section, we commence by providing a simple extension of SSAST with incorporation of waveforms. Subsequently, we delve into the intricate design of our CRAFT framework, discussing its tailored modules in both the pretraining and finetuning stages.

In the high level, CRAFT is equipped with three unique designs: (i) the Multi-Scale Audio Embedding (MSAE), which fills the gap of lacking raw audio waveform embedding in the era of transformer, (ii) the Contrastive Learning between Spectrogram and Waveform representations (SWaCL), which leverages contrastive learning on the natural pairing of audio spectrograms and audio waveforms embedded by MSAE, (iii) the Spectrogram and Waveform representations with fusion Bottlenecks (SWaB), which treats spectrograms and waveforms as different "modalities" and utilizes bottleneck tokens to exchange information between both representations in finetuning.

### 3.1 MSAE: MULTI-SCALE AUDIO EMBEDDING

A pivotal challenge in acoustic modeling revolves around the extraction of frequency-related features from raw waveform representations. Drawing inspiration from prior works in the domain of

raw waveform modeling (Zhu et al., 2018), we design multi-scale feature extraction on waveform embedding technique as Multi-Scale Audio Embedding (MSAE) (the dashed yellow rectangle at the bottom left of Figure. 1). Given an input audio clip $x \in \mathbb{R}^d$, we perform feature extraction at multiple scales, the MSAE operations can be represented as follows:

$$x^s = \theta^s(x), s \in [11, 51, 101] \tag{1a}$$

$$x_{\text{MSAE}} = \text{Concat}([\text{Pool}(x^s)]_s) \tag{1b}$$

$$p_{\text{wav}} = \text{Patchfy}(x_{\text{MSAE}}) \tag{1c}$$

where $\theta^s$ denotes the convolution opreration with a kernel size of $s$ for feature extraction, and $x^s$ signifies the extracted features at different scales accordingly. The 1D convolutions with smaller kernel sizes capture fine-grained temporal-frequency responses, while those with larger kernels extract long-term temporal-frequency characteristics. Sequentially, we employ average pooling and concatenate the results to synchronize the features across different scales, followed by a "Patchfy" operation to split waveform representations into multiple patches similar to spectrogram modeling.

To keep our main contribution more distinct from SSAST, our Patchfy function on both spectrograms and waveforms are the same as SSAST as a convolution operation. Using function "Specify" to denote the conversion from waveforms to spectrograms, the original SSAST embeds spectrogram as: $x \rightarrow \text{Patchfy}(\text{Specify}(x))$. Thus waveforms can be embedded in the similar was as spectrogram: $x \rightarrow \text{Patchfy}(\text{MSAE}(x))$.

### 3.2 SWaPT-SWAST: A NATURAL EXTENSION OF SSAST WITH WAVEFROMS

Closely following the idea of Self-Supervised Audio Spectrogram Transformer (SSAST (Gong et al., 2022a)), a naive extension of it is to treat audio waves as additional inputs, completely separated from the computation of spectrograms. In pretraining, this simple extension of joint Spectrogram and Waveform modeling in PreTraining is named as SWaPT in this paper. In finetuning, the simple extension of waveform is called SWAST, i.e. Spectrogram and Waveform based AST modeling. In summary, we use SWaPT-SWAST to denote the simple extension of SSAST with waveforms added in both pretraining and finetuning. A recap of SSAST can be found in Appendix. A.2.

In pretraining, the SSAST framework applies a masked spectrogram patch modeling (MSPM) objective on audio spectrograms, which consists of both discriminative ($\mathcal{L}_{\text{spec}}^d$) and generative ($\mathcal{L}_{\text{spec}}^g$) training objectives. With $r_g$ to denote the ratio of generative loss, the MSPM loss is denoted as:

$$\mathcal{L}_{\text{MSPM}} = \mathcal{L}_{\text{spec}}^d + r_{\text{g}} \cdot \mathcal{L}_{\text{spec}}^g \tag{2}$$

In SWaPT, we keep spectrogram calculations the same as SSAST and completely isolated from waveforms except the final linear classification layer. For waveform representations, the MSAE-processed tokens are input into the same model backbone, followed by both discriminative and generative training objectives. The training objective on waveforms is the same equation as spectrogram calculations in SSAST except we replace spectrogram tokens $x_{\text{spec}}$ with waveform tokens $x_{\text{wav}}$. The total training loss of SWaPT, denoted as $\mathcal{L}_{\text{SWaPT}}$, is designed to be the sum of spectrogram and waveform losses with a hyper-parameter $r_{\text{w}}$ to control the ratio of loss from waveform representations in the following way:

$$\mathcal{L}_{\text{SWaPT}} = (\mathcal{L}_{\text{spec}}^d + r_{\text{g}} \cdot \mathcal{L}_{\text{spec}}^g) + r_{\text{w}} \cdot (\mathcal{L}_{\text{wav}}^d + r_{\text{g}} \cdot \mathcal{L}_{\text{wav}}^g) \tag{3}$$

In SWAST, we keep every training detail the same as AST, except both spectrograms and waveforms are input to the AST model backbone, and a final linear layer is added to classify the concatenation of the spectrogram outputs and waveform outputs.

While SWaPT-SWAST is a natural extension of SSAST with additional waveform tokens, as shown in Section. 4, we find that SWaPT does not perform comparably well as SSAST. We argue that implicitly, the goal of this simple extension is that the shared model backbone will be capable of extracting separate semantics from both spectrograms and waveforms, simultaneously. However, as discussed in Section. 2, the spectrogram and waveform representations are largely diversified and misaligned, which makes it unreasonable to expect that one shared model backbone achieves strong performances on both spectrograms and waveforms. Consequently, SWaPT and SWAST are not incorporated into the final design of CRAFT, and is only used for ablation study.

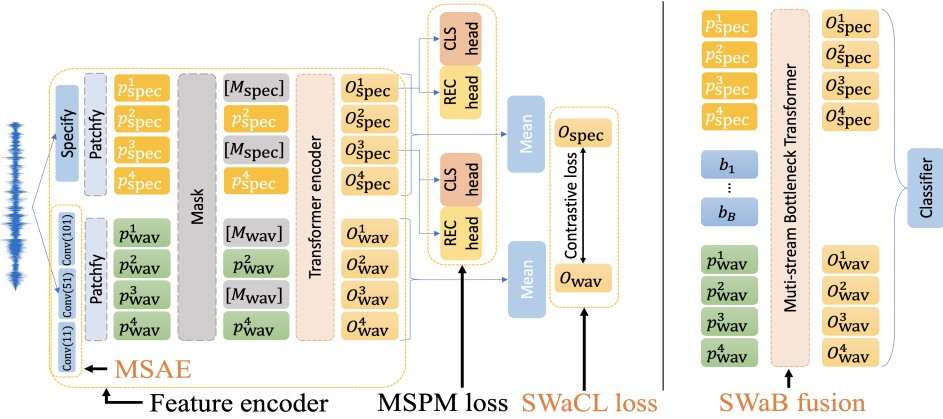

Figure 1: The architecture of proposed CRAFT framework in pretraining (left) and finetuning (right). Both spectrogram and waveform tokens are calculated from the same input audio sample. A spectrogram-waveform contrastive learning loss is incorporated in the pretraining. Bottleneck tokens are added in finetuning to restricts the flow within either spectrogram or wavefrom calculations.

### 3.3 CRAFT IN PRETRAINING: SPECTROGRAM AND WAVEFORM BASED CONTRASTIVE LEARNING (SWACL)

We tackle the semantic misalignment during pretraining, and utilizes contrastive learning to contrast outputs from spectrograms and waveforms after projection onto same high-dimensional space. Our contrastive learning objective is inspired by the simple yet effective framework in SimCLR (Chen et al., 2020a), where the spectrogram and waveform representations naturally constitute the two branches of the SimCLR architecture as plotted in the left of Figure. 1. In SimCLR, contrastive learning is conducted between two branches, typically comprising two augmented views of the same data sample. Thanks to our well-crafted patch embeddings, spectrogram and waveform patches can naturally serve as contrasting pairs, akin to SimCLR. Using the term "Encoder" to denote the common backbone, we denote the post-backbone outputs as $O_{\text{spec}} = \text{Encoder}(p_{\text{spec}})$ and $O_{\text{wav}} = \text{Encoder}(p_{\text{wav}})$. The InfoNCE (Oord et al., 2018) contrastive loss (denoted as $\mathcal{L}_{\text{SWaCL}}$) in SWaCL is defined in Equation. 4a. With $r_{\text{cl}}$ to denote the ratio of contrastive learning loss, the total loss in CRAFT pretraining stage is the joint MSPM loss and SWaCL loss, called PSWaCL loss, and is formulated as Equation. 4b.

$$\mathcal{L}_{\text{SWaCL}} = -\frac{1}{N} \sum_{i=1}^{N} log\left(\frac{exp((O_{\text{wav}}^i)^T O_{\text{spec}}^i)}{\sum_{j=1}^{N} exp((O_{\text{wav}}^i)^T O_{\text{spec}}^j)}\right) \tag{4a}$$

$$\mathcal{L}_{\text{PSWaCL}} = \mathcal{L}_{\text{MSPM}} + r_{\text{cl}} \cdot \mathcal{L}_{\text{SWaCL}} \tag{4b}$$

$$= (\mathcal{L}_{\text{spec}}^d + r_{\text{g}} \cdot \mathcal{L}_{\text{spec}}^g) + r_{\text{cl}} \cdot \mathcal{L}_{\text{SWaCL}} \tag{4c}$$

### 3.4 CRAFT IN FINETUNING: SPECTROGRAM AND WAVEFORM MODELING WITH FUSION BOTTLENECKS (SWAB)

To maximize the collective modeling on spectrograms and waveforms during finetuning, we tackle the temporal misalignment by incorporating fusion bottlenecks as proposed by Nagrani et al. (2021), and thoughtfully design the backbone to treat spectrogram and waveform as distinct modalities. These enhancements serve to bolster the model's performance in downstream tasks.

We introduce Spectrogram and Waveform modeling with fusion Bottlenecks (SWaB). In SWaB, we employ a common model backbone for both spectrogram and waveform representations and introduce additional bottleneck tokens. Treating spectrograms and waveforms as distinct modalities and applying bottlenecks to mitigate temporal misalignment empirically enhance our performances.

In more detail, as shown in the right of Figure. 1, we introduce a small set of fusion bottleneck tokens, denoted as $b_1, ..., b_B$ where $B$ is the number of bottleneck tokens, into the input spectrogram tokens $t_{\text{spec}}$ and waveform tokens $t_{\text{wav}}$. If we opt to apply fusion bottlenecks at one transformer encoder layer, denoted as Encoder$_l$, we pass tokens through Encoder$_l$ twice. During each pass, we

first concatenate the representation tokens with bottleneck tokens and then proceed with attention and other computation operations. Mathematically, the forward passes are represented as follows:

$$[o_{\text{spec}}, b_1^s, ..., b_B^s] = \text{Encoder}_l(\text{Concat}(t_{\text{spec}}, b_1, ..., b_B)) \tag{5a}$$

$$[o_{\text{wav}}, b_1^w, ..., b_B^w] = \text{Encoder}_l(\text{Concat}(t_{\text{wav}}, b_1, ..., b_B)) \tag{5b}$$

$$b_1, ..., b_B = \text{mean}(b_1^s, b_1^w), ..., \text{mean}(b_B^s, b_B^w) \tag{5c}$$

In above equations, Concat denotes concatenation and $o_{\text{spec}}$ / $o_{\text{wav}}$ are the layer outputs, with respect to spectrogram and waveform tokens. The bottlenecks are updated by averaging bottleneck outputs from each representation. This approach ensures that we not only fully leverage the modeling capabilities on both representations but also alleviate the temporal misalignment.

## 4 EXPERIMENTS

### 4.1 DATASETS

We use two commonly used dataset for experiments in this paper:

**AudioSet (Gemmeke et al., 2017)** consists of more than 2 million 10-second audio clips with 527 labels. The AudioSet training set has been split into two sections: full set and balanced set comprising 2M and 22k samples, respectively. We report mean average-precision (mAP) following previous works. **ESC50 (Piczak, 2015)** consists of environmental sound recordings categorized into 50 classes, with each class having 40 samples. To maintain consistency with previous research, we conduct a 5-fold cross-validation and report the average accuracy as our performance metric.

### 4.2 IMPLEMENTATION DETAILS

During pretraining, to maintain consistency with prior research, we largely adopted the training methodology outlined in SSAST (Gong et al., 2022a). Specifically, we first truncate or pad the input audio clip from AudioSet to 10 seconds. For spectrogram calculations, each wave signal is converted into a 128-dimention log Mel filterbank calculated with a 25 ms window size for every 10 ms, and then split into patches of size $16 \times 16$, followed by patch-embedded into a 768-dimensional vector. For waveform calculations, the 10-sec audio clip is input into MSAE with kernel sizes of 11, 51 and 101. During the pretraining process, we utilize a batch size of 180, employ the Adam optimizer (Kingma & Ba, 2015) with an initial learning rate set to 5e-4. With contrastive loss introduced in SWaCL, we set the CL loss ratio as 1e-2 to balance the patch masking loss and contrastive loss.

For downstream tasks, following the same training setting as (Gong et al., 2022a), we use learning rate of 5e-5 and 1e-4, total epoches of 25 and 50, for AudioSet and ESC50, respectively. The mixup (Tokozume et al., 2018) and SpecAugment (Park et al., 2019) is used for data augmentation.

### 4.3 MAIN RESULTS

**AudioSet:** Table. 1 lists the results for AudioSet. We first compare the proposed CRAFT to previous SOTA methods taking either waveform or spectrogram as inputs. CRAFT clearly outperforms the previous works using the waveform input (EnvNet-v2), and works with energy map as input (AST) without ImageNet weights initialization. This demonstrates the effectiveness of proposed pretraining that aligns the tokens of different representations from same acoustic modality. We further compare the CRAFT with SSAST (Gong et al., 2022a), which is the spectrogram-only counterpart of CRAFT as well as one of the previous state-of-the-art SSL works. On AudioSet-bal, our proposed CRAFT achieves higher mAP by a significant margin with 4.4% higher mAP compared with SSAST (from 29.0% to 33.4%), demonstrating the different representations from same modality is complementary to each other. We finally compare the proposed CRAFT with previous works on their best setup, we find that CRAFT is able to achieve SOTA comparable results on Audioset full set, showing that the proposed method is able to fully utilize the information from different representations of same modality. Note that the CRAFT does not rely on the image dataset pretrained weights, which shows that the proposed pretraining is able to effectively learn the acoustic semantics from scratch.

**ESC50**: We show CRAFT performance on ESC50 in Table. 1. CRAFT achieves SOTA comparable performances and clearly outperforms SSAST with 5.4% (from 84.7% to 90.1%) higher accuracy, demonstrating the findings from AudioSet generalize well to small scale dataset.

Table 1: Evaluation on AudioSet and ESC50 dataset. We report mAP for AudioSet and top1 average accuracy for ESC50. IN SL denotes ImageNet pretraining. * denotes larger batch size of 512. And [ens] denotes ensemble of multiple runs.

| Models | Pretrain | Audio inputs | AudioSet-bal % | AudioSet-unbal % | ESC50 % |
|---|---|---|---|---|---|
| **Existing supervised works with audio wave input** | | | | | |
| M18 (Dai et al., 2017) | IN SL | Wav | - | - | 71.7 |
| WaveMsNet (Zhu et al., 2018) | IN SL | Wav | - | - | 79.1 |
| EnvNet-v2 (Tokozume et al., 2018) | IN SL | Wav | - | - | 84.9 |
| **Existing supervised works with energy map input** | | | | | |
| PANN (Kong et al., 2020) | IN SL | Spec | 27.8 | 43.9 | 94.7 |
| AST (Gong et al., 2021) | IN SL | Spec | 34.7 | 45.9 | 95.6 |
| AST (Gong et al., 2021) | - | Spec | 14.8 | 36.6 | 88.7 |
| HTS-AT (Chen et al., 2022) | IN SL | Spec | - | 47.1 | 97.0 |
| CAT (Liu et al., 2023) | IN SL | Spec | - | 47.8 | - |
| **Existing patch masking encoder-decoder works with energy map input** | | | | | |
| MAE-AST (Baade et al., 2022) | SSL | Spec | 30.6 | - | - |
| Audio-MAE* (vanilla) (Huang et al., 2022a) | SSL | Spec | 36.6 | 46.8 | - |
| MaskSpec (Chong et al., 2023) | SSL | Spec | 32.3 | 47.1 | - |
| **Existing patch masking encoder-only works with energy map input** | | | | | |
| 24-SSAST-AST †(Gong et al., 2022a) | SSL | Spec | 29.0 | - | 84.7 |
| 180-SSAST-AST | SSL | Spec | 31.4 | - | 87.9 |
| **Our self-supervised CRAFT with energy map and audio wave inputs** | | | | | |
| SWaPT-AST | SSL | Spec + Wav | 27.4 | - | 87.5 |
| SWaPT-SWAST | SSL | Spec + Wav | 27.7 | - | 84.8 |
| SWaPT-SWaB | SSL | Spec + Wav | 29.4 | - | 85.4 |
| PSWaCL-AST | SSL | Spec + Wav | 31.3 | - | 88.0 |
| SSAST-SWaB | SSL | Spec + Wav | 31.6 | - | 87.7 |
| **CRAFT** | **SSL** | **Spec + Wav** | **33.4** | **43.0 / 46.1**[ens] | **90.1** |

Table 2: Evaluated methods during pretraining and finetuning

| Stages | Methods | Explanations |
|---|---|---|
| Pretrain | SSAST | The original SSAST method (Gong et al., 2022a). |
| | SWaPT | A simple extension of SSAST with wav tokens. |
| | SWaCL | Only use contrastive objective in pretraining. |
| | PSWaCL | Our proposed pretraining method with SWaCL loss. |
| Finetune | AST | The original finetuning method in SSAST. |
| | SWAST | A simple extension of SSAST with wav tokens. |
| | SWaB | Our proposed finetuning method with bottleneck tokens. |

## 4.4 ABLATIONS AND KEY FINDINGS

We conduct ablations on AudioSet balanced set, and use same model and training settings for ablations unless specified. To better assist the explanations, in this paper we adopt a standardized format for each framework, presented as BatchSize-PretrainMethod-FinetuneMethod. For example, the "24-SSAST-AST" is the original SSAST framework proposed in (Gong et al., 2022a), which has batch size of 24, the MSPM training objective in pretraining, and original AST for finetuning. And "180-PSWaCL-SWaB" is our proposed CRAFT framwork with batch size of 180, PSWaCL objectives in pretraining and SWaB bottlenecks in finetuning. Unless specified explicitly, we set the batch size default to 180, expanded from batch size of 24 in SSAST. In total, we conducted comprehensive evaluations with 4 pretraining methods and 3 finetuning methods, listed in Table. 2.

**Cross-representation Fusion: A simple extension of SSAST with waveform representations is not performing as well as the original SSAST.** The SWaPT-SWAST, discussed in Section. 3.2, is the straightforward extension of SSAST-AST, incorporating waveform representations. However, from Table. 1, all SWaPT-pretraining results lag behind other frameworks by a clear margin. This shows that directly aggregating results from waveform based semantics and spectrogram based semantics will actually harm the performance. Due to the distinct input formats of spectrograms and waveforms, the misalignment rooted in both semantic space and temporal granularities can not be

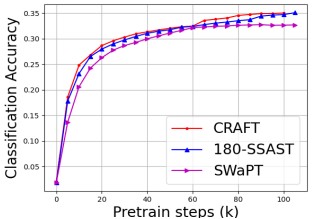 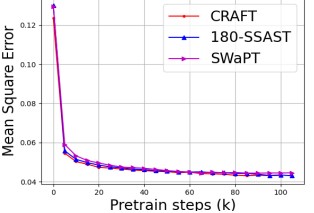 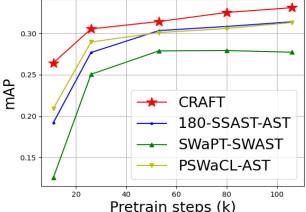

(a) Masked accuracy in pretraining    (b) Masked MSE in pretraining    (c) FT performances vs PT steps

Figure 2: Evaluation metrics of each framework with respect to pretraining steps. During pretraining, we plot prediction accuracy (left) and reconstruction MSE (middle). During finetuning, we plot mAP on AudioSet balanced set (right).

Table 3: Bottleneck tokens in finetuning.

| Frameworks | 180-SSAST-AST | 180-SSAST-SWaB | SWaPT-AST | SWaPT-SWAST | SWaPT-SWaB | PSWaCL-AST | PSWaCL-SWaB |
|---|---|---|---|---|---|---|---|
| mAP % | 31.4 | 31.6 | 27.4 | 27.7 | 29.4 | 31.3 | 33.4 |

improved in SWaPT-SWAST. In contrast, by applying the proposed pretraining with SWaCL loss and the finetuning with bottleneck tokens, we observe a 4.4% performance gain in CRAFT.

**Semantic Alignment: The SWaCL loss between spectrograms and waveforms presents a challenging yet not insurmountable pretraining objective.** To illustrate the performances during pretraining, we present pretraining metrics, including masked classification accuracy and masked Mean Squared Error loss, as well as finetuning metrics, i.e. mean Average Precision (mAP), in Figure. 2. While PSWaCL employs additional contrastive learning, we observe that the pretraining metrics for 180-SSAST and PSWaCL are close to each other. This suggests that as opposed to the naive incorporation of waveforms in SSAST, the addition of contrastive loss, which mititages the semantic alignment, will not hinder the pre-training performance of the spectrogram representation. In consequence, during the fine-tuning phase, the features from waveform representation complements the energy based representation and leads to better performances (red curve in Figure. 2c).

**Temporal Alignment: The temporal alignment between spectrogram and waveform leads to a notable enhancement, especially when pretrained on both spectrogram and waveform representations.** Comparing AST finetuning and SWaB finetuning, i.e. PSWaCL-AST vs PSWaCL-SWaB and SWaPT-SWAST vs SWaPT-SWaB, we observe around 2.0% mAP increase in Table. 3 when bottlenecks are applied during finetuning. This demonstrates that spectrograms and waveforms encompass complementary semantic information, and bottlenecks effectively mitigate temporal misalignment during finetuning by restricting information exchange. This indicates that similar to video understanding, temporal alignment is also critical for audio understanding tasks such as classification. However, it's worth noting that SSAST-AST and SSAST-SWaB exhibit similar performances. We posit that this is because the model is not exposed to any waveform representations during SSAST pretraining, and therefore temporal alignment during finetune brings little to no benefits.

Table 4: Larger batch sizes benefits SSAST and CRAFT.

| Models | 24-SSAST-AST | 180-SSAST-AST | 24-PSWaCL-AST | 24-PSWaCL-SWaB | 180-PSWaCL-AST | 180-PSWaCL-SWaB |
|---|---|---|---|---|---|---|
| mAP % | 29.0 | 31.4 | 17.6 | 13.9 | 31.3 | 33.4 |

**Batch-size Matters: Larger batch sizes significantly boost the performances for contrastive learning and SSAST.** The original SSAST was trained with a small batch size of just 24, which poses limitations for contrastive learning. Small batch sizes restrict the number of negative samples within the same batch, making the contrastive learning (CL) objective more challenging in distinguishing positive from negative samples. We ablate the batch size and show results in Table 4. Similar to the previous findings from computer vision, the larger batch size will lead to better performances especially for PSWaCL, because with more negative samples within each batch, the model is able to contrast and capture distinct semantics from both spectrogram and waveform reprsentations more effectively. By increasing the batch size from 24 to 180, CRAFT doubles its performances.

**Ablation on Modules.** We ablate the usage of carefully designed building modules of SWaCL and SWaB in CRAFT in Table. 5a. Starting from the original SSAST with 29.0%, larger batch size with tuned learning rate enables 2.4% higher mAP. Further adding contrastive loss SWaCL, the downstream performancs remain the same. Next, incorporating SWaB can improve mAP by

Table 5: Ablations results.

(a) SWaCL and SWaB.

| Modules | mAP % |
|---|---|
| SSAST | 29.0 |
| + Larger batch | + 2.4 |
| + SWaCL | - 0.1 |
| + SWaB | + 2.1 |

(b) Multi-scales in MSAE.

| Pre-train | mAP % |
|---|---|
| Single-scale | 31.5 |
| **Multi-scale** | **33.4** |

(c) # bottlenecks.

| # bottlenecks | mAP % |
|---|---|
| 4 | 33.3 |
| 16 | 33.1 |
| **64** | **33.4** |
| 128 | 32.6 |

(d) Starting fusion layer.

| Fusion layer | mAP % |
|---|---|
| 0 | 33.1 |
| **4** | **33.4** |
| 8 | 32.8 |
| 12 | 33.0 |

another 2.1%. In total, CRAFT achieves 4.4 % higher mAP compared with the spetrogram-only counterpart. Together with observations in Table. 3, we find that either PSWaCL-only approach or SWaB-only approach can only match the performances of SSAST, and best results of CRAFT can only be achieved with joint PSWaCL and SWaB modules.

**Ablation on multi-scale embeddings in MSAE.** To demonstrate that multi-scale embeddings help capture multiple levels of semantics and benefit our final performances, we compare the results of the proposed MSAE in Section. 3.1 and the single scale waveform embedding with one scale $s = 51$ in Equation. 1a. The results are given in Table. 5b. We observe that three-scale input embedding works much better than single scale with 1.9% higher mAP. This shows the effectiveness of our proposed convolution based feature extraction on raw waveforms. Since we would like to make this work focus on the cross-representation modeling, we do not explore even more scales in MSAE, and leave advanced waveform embeddings as future work.

**Ablation on number of bottleneck tokens and fusion layers during finetuning.** As shown in the original MBT paper (Nagrani et al., 2021), only a small number of bottleneck tokens are sufficient for cross-modality fusion. The smaller number of bottleneck tokens, the less information will be exchanged and the more calculation will be restricted within each modality, and vice versa. We varied the number of bottleneck tokens, and obtained similar behaviors shown in Table. 5c as MBT. We notice that less than 64 fusioin tokens are sufficient to fuse representations from two streams and achieve a higher mAP than 128 tokens. We continue to ablate the starting fusion layer of bottleneck fusion. The exploration of the location of fusions also involves the comparison of early, middle and late fusion, similar as the vision domain (Jaegle et al., 2021; Nagrani et al., 2021). The ablation results are given in Table. 5d, where we find that middle fusion starting from layer 4 achieves the best performances. Middle fusion outperforms early fusion (i.e. fusion starting from layer 0) and late fusion (i.e. fusion starting from layer 12) by 0.4% mAP.

While this a pioneering work in audio classification that successfully bolstered self-supervised audio modeling after incorporating waveform representations, we would like to list several interesting discussions. First, the performance boost from waveform is not free food, which doubles the computation resources since there are two forward passes from spectrogram and waveform calculations. This leads us to explore that spectrogram and waveform modalities may benefit from distinct backbone architectures, and consider the search of a light-weight backbone tailored for either spectrogram or waveform as a future work. Second, while CRAFT successfully boosts MSPM to match the performances of Masked AutoEncoder (MAE) based methods (Audio-MAE (Huang et al., 2022a)), the combination of CRAFT and MAE is interesting and promising. Recently, It has been validated in Huang et al. (2022b); Gong et al. (2022b); Mishra et al. (2023) that contrastive learning and Masked Auto-Encoder are complementary. The masked encoder-decoder architecture from MAE presents a powerful patch modeling method compared with MSPM and has the potential to further enhance the performances together with SWaCL loss between spectrograms and waveforms.

## 5 CONCLUSION

In this paper, we introduce Cross-Representation modeling on Audio waveForms and specTrograms (CRAFT) framework, an innovative cross-representation modeling in the audio domain. While the majority of prior audio classification methods rely on either frequency-domain spectrograms or time-domain waveforms, CRAFT aligns both representations and successfully enhances the downstream performances. To tackle semantic alignment, CRAFT contrasts spectrograms and waveforms and proposes a novel SWaCL contrastive objective in pretraining. To mitigate the temporal misalignment, SWaB is introduced by adding additional bottleneck tokens in finetuning. In consequence, superior performances can be achieved in downstream tasks. Future works involve the incorporation of masked autoencoder based methods and the optimization of computation consumptions.

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

## A APPENDIX

### A.1 EXTRA ABLATIONS

**SWaCL and MSPM objectives in CRAFT are complementary to each other.** To validate the relationship between contrastive loss and the original MSPM mechanism, we concudcted several experiments, namely SWaCL-AST and SWaCL-SWaB, where only contrastive loss is used in pretraining. As we can see from Table. 6, the downstream performances of SWaCL-AST are significantly degraded, with only around 12% mAP. Neither MSPM-only pretraining nor SWaCL-only pretraining achieve the best performances.

Table 6: SWaCL and MSPM in CRAFT are complementary.

| Models | SSAST-AST | SSAST-SWaB | SWaCL-AST | SWaCL-SWaB | PSWaCL-AST | PSWaCL-SWaB |
|---|---|---|---|---|---|---|
| mAP % | 31.4 | 31.6 | 11.1 | 12.8 | 31.3 | 33.4 |

**Impact of CL loss ratio in PSWaCL during pretraining.** Incorporating contrastive learning into CRAFT's pretraining introduces a hyperparameter: the ratio of contrastive loss (CL) in the total loss, i.e. $r_{cl}$ in Equation. 4. Through empirical experimentation, we have determined that a CL ratio of 0.01 yields favorable results in PSWaCL. Deviating from this optimal ratio, either by setting it too high or too low, has negative consequences on downstream task performances. For instance, when using a small CL ratio, the downstream task exhibits worse performance (i.e., 0.6% worse mAP when CL ratio = 0.0001), and in extreme cases, setting the CL ratio too high can lead to catastrophic performance collapse (e.g., only achieving around 10% mAP when CL ratio = 1).

**Impact of model sizes.** In all frameworks evalualted in this work, we use AST model architecture as the modeling backbone. By default, we report the performances of base model as proposed in AST. Moreover, we also test the performances of two model size variants, i.e. small and tiny models.

For each model architecture, we show the results on AudioSet balanced set in Table. 7. We find that CRAFT scales well with larger model sizes, and large models significantly boost the performances from 20.6% up to 33.4%.

Table 7: Ablation of different model sizes.

| Models | AudioSet-bal % | ESC50 % |
|---|---|---|
| CRAFT (T/S/B) | 20.6/27.5/33.4 | 73.7/84.0/90.1 |

## A.2 RECAP OF SSAST FRAMEWORK

The SSAST framework operates on audio spectrograms only, employing a masked spectrogram patch prediction (MSPM) framework to enhance AST pretraining for downstream tasks. MSPM initially masks a portion of audio patches, allowing for the controlled clustering of masked patches through a cluster factor. Subsequently, MSPM utilizes both discriminative and generative training objectives to improve the pretraining model's performance across audio and speech tasks.

To elaborate further, given one input audio waveform $x \in \mathrm{R}^l$, we will first calculate its spectrogram (denoted as function "Specify"), then split it into patches $p_{\text{spec}}$ (denoted as function "Patchfy").

$$x_{\text{spec}} = \text{Specify}(x) \tag{6a}$$

$$p_{\text{spec}} = \text{Patchfy}(x_{\text{spec}}) \tag{6b}$$

Following this, we generate a set $I$ comprising $N$ masked patch position indexes. For each patch with an index belonging to $I$, its patch embedding is replaced with a learnable mask embedding $E_{\text{mask}}$. Subsequently, both masked and unmasked patches are fed into a transformer encoder, yielding encoded outputs $O_{\text{spec}}^i$. Equipped with both discriminative and generative training objectives, $O_{\text{spec}}^i$ was then input to a classification head and reconstruction head, and two outputs $c_{\text{spec}}^i$ and $r_{\text{spec}}^i$ were obtained, respectively. To optimize the discriminative objective, the InfoNCE (Oord et al., 2018) loss $\mathcal{L}_{\text{spec}}^d$ is employed, along with the mean square error (MSE) loss $\mathcal{L}_{\text{spec}}^g$ for the generative objective. The InfoNCE loss matches $(x_{\text{spec}}^i, c_{\text{spec}}^i)$ pairs for discriminative objective, and the MSE error approximates $x_{\text{spec}}^i$ with $r_{\text{spec}}^i$ for generative objective. The total loss on spectrogram tokens is then a weighted sum of $\mathcal{L}_{\text{spec}}^d$ and $\mathcal{L}_{\text{spec}}^g$, denoted as $\mathcal{L}_{\text{spec}}$ with a weighting factor of $r_{\text{g}}$. MSPM introduced a challenging yet possible objective for self-supervised audio task in pretraining stage, thus enhancing the model in downstream audio and speech tasks.

$$\mathcal{L}_{\text{spec}}^d = -\frac{1}{N} \sum_{i=1}^{N} log(\frac{exp((c_{\text{spec}}^i)^T x_{\text{spec}}^i)}{\sum_{j=1}^{N} exp((c_{\text{spec}}^i)^T x_{\text{spec}}^j)}) \tag{7a}$$

$$\mathcal{L}_{\text{spec}}^g = \frac{1}{N} \sum_{i=1}^{N} (r_{\text{spec}}^i - x_{\text{spec}}^i)^2 \tag{7b}$$

$$\mathcal{L}_{\text{spec}} = \mathcal{L}_{\text{spec}}^d + r_{\text{g}} \cdot \mathcal{L}_{\text{spec}}^g \tag{7c}$$

In the finetuning of SSAST, loaded with SSAST-pretrained checkpoint, the original AST model was utilized with the same implemntation details such as learning rate, weight decay, etc. To better clarify the framework, we refer to the original SSAST framework as SSAST-AST, meaning SSAST used in preptraining and AST used in finetuning.

## A.3 WAPT-AST: REPLACE SPECTROGRAMS WITH WAVEFORMS IN SSAST

To validate our unique contributions on contrastive learning on multi-representations, we ablate the representation modeling from spectrograms to waveforms in this section. And we will call the Waveform-based PreTraining and AST finetuning as WaPT-AST.

As explained in Section. 3.1, to construct waveform embedding, we designed MSAE modeling. Thus the spectrogram embedding in the original SSAST-AST is denoted as Equation. 8. The only difference in WaPT-AST is to replace it with Equation. 9.

$$x \to \text{Patchfy}(\text{Specify}(x)) \tag{8}$$

$$x \rightarrow \text{Patchfy}(\text{MSAE}(x)) \tag{9}$$

Table 8: Ablations of single-representation learning.

|  | SSAST-AST | WaPT-AST | PSWaCL-SWaB |
|---|---|---|---|
| AudioSet-bal mAP % | 29.0 | 24.4 | 33.4 |
| ESC50 acc % | 84.7 | 82.0 | 90.1 |

The results of WaPT-AST is given in Table. 8. Our proposed CRAFT, i.e. PSWaCL-SWaB, successfully boosted the performances on both AudioSet and ESC50 datasets. After carefully grid-searched hyper-parameters, WaPT still has a 9.0% smaller mAP than CRAFT. Moreover, we noticed that the AST modeling on spectrograms is largely improved over modeling on waveforms, i.e. 4.6% higher mAP on AudioSet-bal set.

