# OpenReview forum: "CRAFT: Cross-Representation modeling on Audio waveForms and specTrograms"
_ICLR.cc/2024/Conference — Submitted to ICLR 2024_

### Official Review · Reviewer_aBV8 · 2023-10-29

**Soundness:** 1 poor
**Presentation:** 1 poor
**Contribution:** 2 fair
**Rating:** 3
**Confidence:** 5

**Summary:**

The authors investigate the use and fusion of two common feature representations within the audio domain, the raw waveform and spectrogram.

**Strengths:**

The topic addressed in the paper is interesting.

**Weaknesses:**

The introduction is poorly written. There are too many terms introduced e.g., PSWaCL, SWaB, MSAE without details of what to expect in the paper and what are the real contributions of the paper.
Challenges in feature fusion are not clear, and a lot of statements are loose or vague.
--> waveform-based features concentrate more on capturing common patterns! What are these common patterns?
--> Spectrograms predominantly emphasize time-frequency responses! What are these responses? Are we assuming a system here? Why can't waveform learning using a learned filterbank do the same?
--> Enhance the comprehensiveness of the feature set! Comprehensiveness in what sense?
None of the above-mentioned statements are actually related to Semantic Misalignment. Semantic means something else.
Temporal Misalignment: Again, the claim by the authors is wrong. In both cases, linear/non-linear processing methods are available and temporal alignment can be easily achieved. It's basic DSP! Nevertheless, One can always do late fusion after learning complementary features.

Related works: There is no mention of existing approaches that have tried feature fusion, which should be the main focus.  Instead, authors have just discussed existing approaches for audio classification, which could be omitted or briefly mentioned if compared against in the experimental section.
https://dcase.community/documents/challenge2021/technical_reports/DCASE2021_Fedorishin_97_t1.pdf
https://www.mdpi.com/1424-8220/19/7/1733
https://www.isca-speech.org/archive/pdfs/interspeech_2018/yang18c_interspeech.pdf
https://dl.acm.org/doi/pdf/10.1145/3240508.3240631

Method (Sec3): Overall, the proposed method is just a combination of well-known existing methods and small extension of  method by Gong et al., 2022a. Novelty is limited and not well highlighted in the context of the problem addressed in the paper.

--> Our work is built upon SSAST. What is SSAST?
--> fills the gap of lacking raw audio waveform embedding in the era of transformer. Again, this is a loose statement that is not explained.
--> Contrastive learning is widely used in multimodal generative models. So, the method is not novel in itself. What do we mean by natural or unnatural pairing?
--> MSAE is a known technique for designing adaptively learned filter banks. Whats novel here? Authors should refer to existing works here.
Patichyfy operation is not explained. A diagram would help readers. Pooling will reduce the information for short kernel-based conv outputs with bigger dimensions. Instead, zero padding, dilation, adaptive strides, and deformed convolution kind of ideas can be used to learn multiscale features. In current practice, pooling has been established to be one of the worst choices.
--> what is specify in Fig1?
--> There is no description of how spectrogram and waveform feature inputs are processed in the transformer frontend. Is it a single transformer with shared weights or individual ones? A lot of these crucial details are superficially treated.
--> spectrogram and waveform patches can naturally serve as contrasting pairs. It is unsure how this will happen. Is there a ref to existing work to establish this?
--> what is t_spec t_wav in (5a,5b)? what are the dimensions? on which axis is the concatenation is happening? Again, it is unclear from where bottleneck features will come? Why are they required? Can't we just project the features to the same dimensional space? What's the design of a multi-stream bottleneck transformer?

Authors have used Mel-spec, which is a non-linear feature, and the arguments in the introduction about miss-alignment due to fixed resolution are in contrast. While I understand the author's point of view, the way things are explained or presented is misleading for a wider audience.
Only spectral domain augmentation is used. Why not the time domain? Existing works have utilized both for acoustic modelling and feature fusion using CNN backbone with remarkable success. Yes, transformers are hot these days! but they only shine for sequence modelling tasks. For classification CNNs are still the best (attention can be incorporated, too); one has just to train them well.


Experimental results are not SOTA, and strong baselines are not considered.
Authors are encouraged to see https://paperswithcode.com/sota/audio-classification-on-esc-50
Existing works have achieved over 98% on ECS-50 benchmark.

Similarly, for Audioset: https://paperswithcode.com/sota/audio-classification-on-audioset



The code is not available to replicate main experiments without which the claims hold no value at venues like ICLR.

**Questions:**

Please see the detailed feedback above.

---

> ### Author Response · Authors · 2023-11-21
> **Response to reviewer aBV8 [Part 1/4]**
>
> Dear Reviewer aBV8,
>
> Thank you so much for taking the time to read our paper and providing very valuable comments and suggestions regarding the novelty and presentations. Please see our point-by-point responses.
>
> > The introduction is poorly written. There are too many terms introduced e.g., PSWaCL, SWaB, MSAE without details of what to expect in the paper and what are the real contributions of the paper.
>
> Thank you for the valuable suggestion. We have carefully revised our introduction section to give a brief explanation about our proposed MSAE, SWaPT, PSWaCL and SWaB.
>
> > Challenges in feature fusion are not clear, and a lot of statements are loose or vague. --> waveform-based features concentrate more on capturing common patterns! What are these common patterns? --> Spectrograms predominantly emphasize time-frequency responses! What are these responses? Are we assuming a system here? Why can't waveform learning using a learned filterbank do the same?
>
> Regarding the challenges of feature fusion, we aim to differentiate the focus of spectrogram and waveform representations. Spectrograms and waveforms are more focused on time-frequency and time-amplitude features, respectively. While acknowledging that we can absolutely apply a learned filterbank to the waveforms, our primary goal is not to answer the questions, such as whether waveform can replace spectrogram, or how to determine the best representation from spectrogram or waveform. Instead, our unique contribution is a novel contrast learning based method to combine complementary information from spectrogram and waveform.
>
> > --> Enhance the comprehensiveness of the feature set! Comprehensiveness in what sense?
>
> It has been established in various audio papers ([5-9]) that, waveforms and spectrograms convey complementary information. By combining waveform and spectrogram, the model classifies the audio input according to a combined feature set, instead of spectrogram only or waveform only.
>
> > None of the above-mentioned statements are actually related to Semantic Misalignment. Semantic means something else. Temporal Misalignment: Again, the claim by the authors is wrong. In both cases, linear/non-linear processing methods are available and temporal alignment can be easily achieved. It's basic DSP! Nevertheless, One can always do late fusion after learning complementary features.
>
> By the discussion of semantic and temporal misalignment, we aim to differentiate the design of CRAFT during pretraining and finetuning, respectively. During pretraining, we designed the model to contrast spectrogram-waveform representations. To do so, the transformer backbone first generates global representations on spectrogram and waveform, on which the contrastive loss is conducted. Only final outputs from different representations are contrasted, which helps alleviate the global semantics misalignment. During finetuning, we intentionally designed the model to exchange cross-representation information at each transformer layer. Due to the different temporal granularities, fusion bottlenecks were utilized to restrict most of attention flow within each modality. In contrast to pretraining, the active information exchange at each layer helps mitigate the temporal misalignment.
>
> While acknowledging that we can always do late fusion on spectrogram and waveform, as demonstrated by SWaPT-SWAST in Table. 1, the extension of simply concatenating waveform and spectrogram by late fusion in both pretraining and finetuning, will actually hurt the performances, from 29.0 mAP to 27.7 mAP. In addition, in ablation Table. 5, the optimal performance of CRAFT is not achieved by late fusion, neither. Therefore, to enhance the cross-representation modeling, we tailored our design to contrast multi-representations in pretraining and restrict cross-representation calculations in finetuning.
>
> > Related works: There is no mention of existing approaches that have tried feature fusion, which should be the main focus. Instead, authors have just discussed existing approaches for audio classification, which could be omitted or briefly mentioned if compared against in the experimental section.
>
> Thank you for the constructive suggestions. We have carefully revised the related work section, and added more audio fusion works to it.
>
>
> > Method (Sec3): Overall, the proposed method is just a combination of well-known existing methods and small extension of method by Gong et al., 2022a. Novelty is limited and not well highlighted in the context of the problem addressed in the paper.
>
> While acknowledging that many techniques used in CRAFT were proposed before, we brought unique insights into the cross-representation audio classification. That is, by treating spectrograms and waveforms as data augmentations to each other in contrastive learning, existing single representation learning can be further enhanced. We believe this is a non-trivial finding and have rephrased our main novelty in the paper.

---

> ### Author Response · Authors · 2023-11-21
> **Response to reviewer aBV8 [Part 2/4]**
>
> > --> Our work is built upon SSAST. What is SSAST?
>
> We added a brief introduction of SSAST in Appendix. A.2.
>
> > --> fills the gap of lacking raw audio waveform embedding in the era of transformer. Again, this is a loose statement that is not explained.
>
> To the best of our knowledge, since the introduction of transformer modeling in audio classification from AST paper, methods based on transformer have shown remarkable performances. However, there is no standard way of embedding waveform representations in transformer. In CRAFT, our goal is not to design an optimal waveform embedding. Instead, we only borrow the muti-scale embedding from previous works about CNN modeling on waveforms.
>
> > --> Contrastive learning is widely used in multimodal generative models. So, the method is not novel in itself. What do we mean by natural or unnatural pairing?
>
> While acknowledging that contrastive learning (CL) is a widely used approach, we argue that applying CL on multi-representations is an non-trivial finding. Typical contrastive learning approaches require tailored data augmentations. For example, in SimCLR paper, authors showed that "composition of data augmentations plays a critical role in defining effective predictive tasks". Our main novelty is that **without tailored designs, spectrograms and waveforms can naturally serve as data augmentations to each other in contrastive learning**. We believe we are the first to do so, and natural spectrogram-waveform is already an additional contrastive pair, which eliminates the necessity of hand-crafted techniques as discussed in SimCLR, such as Cutout, Sobel filtering, etc.
>
> > --> MSAE is a known technique for designing adaptively learned filter banks. Whats novel here? Authors should refer to existing works here.
>
> Thank you for the valuable suggestion. We do not claim MSAE as our main contribution, and just borrow the idea for waveform embedding in the era of transformer. We have added the existing works in Section. 3.1.
>
> > Patichyfy operation is not explained. A diagram would help readers.
>
> To keep our main contribution more clear and reduce the difference between SSAST and CRAFT, our Patchfy function on both spectrograms and waveforms are the same as SSAST as a simple convolution operation. We have revised the explanations accordingly in Section. 3.
>
> > Pooling will reduce the information for short kernel-based conv outputs with bigger dimensions. Instead, zero padding, dilation, adaptive strides, and deformed convolution kind of ideas can be used to learn multiscale features. In current practice, pooling has been established to be one of the worst choices.
>
> We would like to thank you for all the possible solutions. As reviewer aBV8 has mentioned, all these solutions might have superior performances over simple pooling. We would like to explain that we do not intentionally design the downsampling techniques, and just follow the same pooling operations as in [1]. On the other hand, we only claim the waveform embedding as a minor contribution, and will leave more advanced waveform embeddings as future research.
>
> > --> what is specify in Fig1? --> There is no description of how spectrogram and waveform feature inputs are processed in the transformer frontend. Is it a single transformer with shared weights or individual ones? A lot of these crucial details are superficially treated.
>
> The function "Specify" denotes the conversion from waveforms to spectrograms, and has been further explained in the revised Section. 3.1.
> There are multiple reasons why we intentionally use the same transformer on both spectrograms and spectrograms. Firstly, by keeping other factors the same, this makes our contributions more distinct. Secondly, according to our responses to reviewer Cffs, the performance difference between sharing the same backbone (i.e. CRAFT) and having 2 individual backbones (i.e. SWa2-AST), is quite small. At last, sharing the same backbone also reduces the memory consumption.
>
> > --> spectrogram and waveform patches can naturally serve as contrasting pairs. It is unsure how this will happen. Is there a ref to existing work to establish this?
>
> There is no existing work to establish this. Instead, **we claim the contrast between spectrogram and waveform patches as our unique contribution**. As explained before, without tailor-designed data augmentations, both representations themselves already form a natural contrastive pair.

---

> ### Author Response · Authors · 2023-11-21
> **Response to reviewer aBV8 [Part 3/4]**
>
> > --> what is t_spec t_wav in (5a,5b)? what are the dimensions? on which axis is the concatenation is happening? Again, it is unclear from where bottleneck features will come? Why are they required? Can't we just project the features to the same dimensional space? What's the design of a multi-stream bottleneck transformer?
>
> As discussed in Section. 3.4, $t_{spec}$ and $t_{wav}$ are input tokens of one transformer layer. And they have dimension of $[b, l, d]$, where $b$ is the batch size, $l$ is the number of $t_{spec}$ or $t_{wav}$ tokens, and $d$ is the hidden dimension. The concatenation is happening along the token dimension, i.e. $l$. We mainly followed the idea of bottleneck fusion from MBT paper [2], where a few extra bottleneck tokens are added into each modality on each layer.
> The necessity of fusion bottlenecks are:
> 1. While naively concatenating spectrogram and waveform tokens indeed improved the results, we found that it only brought very limited benefits. As in Table. 3, comparing SWaPT-AST and SWaPT-SWAST, we kept pretraining as the same but only changed the finetuning as joint training on spectrogram and waveform instead of training on spectrogram-only. As a result, SWaPT-SWAST has only 0.3 higher mAP compared with SWaPT-AST.
> 2. Due to the largely diversified misalignment between spectrogram and waveform, they can be treated as multi-modal inputs, which naturally brings the usage of fusion bottlenecks as introduced in MBT to solve multi-modal fusion problem.
>
> Again to keep our main contribution more distinct from SSAST, we designed the multi-stream bottleneck transformer the same as the original SSAST. The only difference is that bottleneck transformer takes all spectrogram tokens, waveform tokens and bottleneck tokens as inputs.
>
>
> > Authors have used Mel-spec, which is a non-linear feature, and the arguments in the introduction about miss-alignment due to fixed resolution are in contrast. While I understand the author's point of view, the way things are explained or presented is misleading for a wider audience. Only spectral domain augmentation is used. Why not the time domain?
>
> We understand the reviewer's concern about the data augmentation. The reason why augmentation is not applied on waveform representations is because **waveform itself is treated as data augmentation against spectrograms**. The reason we introduced waveform into SSAST is because we believe contrastive learning on same domain representations can be utilized to further enhance the performances.
> Therefore, treated as one kind of data augmentation, waveform might not need extra augmentation techniques. And we will leave it as a future work. However, as established in CRAFT, contrastive learning on multi-representations can already enhance the audio performances **even if waveform is still un-augmented**.
>
> > Existing works have utilized both for acoustic modelling and feature fusion using CNN backbone with remarkable success. Yes, transformers are hot these days! but they only shine for sequence modelling tasks. For classification CNNs are still the best (attention can be incorporated, too); one has just to train them well.
>
> > Experimental results are not SOTA, and strong baselines are not considered. Authors are encouraged to see https://paperswithcode.com/sota/audio-classification-on-esc-50 Existing works have achieved over 98% on ECS-50 benchmark. Similarly, for Audioset: https://paperswithcode.com/sota/audio-classification-on-audioset
>
> > The code is not available to replicate main experiments without which the claims hold no value at venues like ICLR.
>
> We totally agree with the reviewer that transformer shines for sequence modeling and CNNs are still the best for various classification tasks. We would like to note that our cross-representation techniques on SSAST successfully improved SSAST's performance, which is orthogonal to the backbone design. And we will leave advanced backbone design as a future work, such as CNN model from PANNs [10] and transformer model from HTS-AT [11].
>
> Thank you for pointing out several SOTA results. We agree that CRAFT is SOTA comparable on AudioSet and ESC-50. As explained above, we kept many details the same as our baseline, such as the spectrogram calculations and the multi-stream transformer backbone architecture in SSAST. However, we note that our proposed techniques are especially unique in the sense that we are the first to contrast different representations from the same data, and intentionally treat waveform as data augmentation against spectrogram. Our method can be possibly incorporated into many audio modeling approaches, such as Audio-MAE [3], CAV-MAE [4].
>
> Our source codes, training configurations and model checkpoints are undergoing internal inspection, which will be released upon completion.

---

> ### Author Response · Authors · 2023-11-21
> **Response to reviewer aBV8 [Part 4/4]**
>
> [1] Zhu, Boqing, et al. "Learning environmental sounds with multi-scale convolutional neural network." 2018 International Joint Conference on Neural Networks (IJCNN). IEEE, 2018.
>
> [2] Nagrani, Arsha, et al. "Attention bottlenecks for multimodal fusion." Advances in Neural Information Processing Systems 34 (2021): 14200-14213.
>
> [3] Huang, Po-Yao, et al. "Masked autoencoders that listen." arXiv preprint arXiv:2207.06405 (2022).
>
> [4] Gong, Yuan, et al. "Contrastive audio-visual masked autoencoder." arXiv preprint arXiv:2210.07839 (2022).
>
> [5] Li, Xinyu, Venkata Chebiyyam, and Katrin Kirchhoff. "Multi-stream network with temporal attention for environmental sound classification." arXiv preprint arXiv:1901.08608 (2019).
>
> [6] Fedorishin, Dennis, et al. Investigating waveform and spectrogram feature fusion for acoustic scene classification. Technical report. Detection, Classification of Acoustic Scenes, and Events 2021.
>
> [7] Su, Yu, et al. "Environment sound classification using a two-stream CNN based on decision-level fusion." Sensors 19.7 (2019): 1733.
>
> [8] Yin, Yifang, Rajiv Ratn Shah, and Roger Zimmermann. "Learning and fusing multimodal deep features for acoustic scene categorization." Proceedings of the 26th ACM international conference on Multimedia. 2018.
>
> [9] Yang, Zixiaofan, and Julia Hirschberg. "Predicting Arousal and Valence from Waveforms and Spectrograms Using Deep Neural Networks." Interspeech. 2018.
>
> [10] Kong, Qiuqiang, et al. "Panns: Large-scale pretrained audio neural networks for audio pattern recognition." IEEE/ACM Transactions on Audio, Speech, and Language Processing 28 (2020): 2880-2894.
>
> [11] Chen, Ke, et al. "HTS-AT: A hierarchical token-semantic audio transformer for sound classification and detection." ICASSP 2022-2022 IEEE International Conference on Acoustics, Speech and Signal Processing (ICASSP). IEEE, 2022.

---

### Official Review · Reviewer_7CEx · 2023-11-01

**Soundness:** 2 fair
**Presentation:** 2 fair
**Contribution:** 2 fair
**Rating:** 3
**Confidence:** 2

**Summary:**

The paper proposed a modeling approach that incorporates both waveform and spectrogram features in the audio domain. The authors also addressed the semantic misalignment and temporal alignment issue raised by the combination. The experiments demonstrate the effectiveness of the approach in audio classification tasks.

**Strengths:**

- Overall, it's an interesting idea to combine both waveform and spectrogram features and address alignment issues in one shot.
- The experiments and ablation studies are quite compressive.

**Weaknesses:**

- Novelty is limited considering the ICRL standards. It might be a better fit for speech-related conferences (e.g. Interspeech)
- The writing of this paper needs to be improved. Too many acronyms to make it less readable and hard to follow the idea, especially in section 3.2.

**Questions:**

In table 1, it shows that the performance of AST (spectrum-only) is still better than all the proposed methods in the paper. How to explain it?

---

> ### Author Response · Authors · 2023-11-21
> **Response to reviewer 7CEx**
>
> Dear Reviewer 7CEx,
>
> Thank you so much for taking the time to read our paper and providing very valuable comments and suggestions regarding the novelty and presentations. Please see our point-by-point responses.
>
> > Novelty is limited considering the ICRL standards. It might be a better fit for speech-related conferences (e.g. Interspeech)
>
> While acknowledging that many techniques in our paper are present in the current deep learning community, we would like to highlight that proposed in CRAFT, the **contrastive learning between different representations from same data** is a non-trivial contribution. In order to learn complementary information from different representations, we selected other techniques and carefully designed each component in our framework.
>
> We would like to note that our contrastive learning on multi-representations can be possibly extended to other domains as well, and we mainly tested it on audio tasks as a showcase. Thus we believe CRAFT might be a good fit for audio representation learning in ICLR.
>
> > The writing of this paper needs to be improved. Too many acronyms to make it less readable and hard to follow the idea, especially in section 3.2.
>
> Thank you for your suggestions on writings. We have revised the acronyms, and added a recap of SSAST in our revision for better presentation.
>
> > In table 1, it shows that the performance of AST (spectrum-only) is still better than all the proposed methods in the paper. How to explain it?
>
> This is a good finding that AST on spectrum-only can beat our results. We would like to mention that AST is a supervised learning method with labels available, and our approach is based on self-supervised setting where labels are missing. Generally in audio classification, supervised learning approaches have a clear advantage over self-supervised learning. For example, AST and SSAST have 34.7% and 29.0% mAP on AudioSet-bal set, which means a gap of 5.7%. However, our proposed CRAFT achieved a mAP of 33.4%, **significantly reduced the gap from 5.7% mAP to 1.3% mAP**.

---

### Official Review · Reviewer_Cffs · 2023-11-01

**Soundness:** 3 good
**Presentation:** 2 fair
**Contribution:** 3 good
**Rating:** 5
**Confidence:** 4

**Summary:**

The paper proposes a joint spectrogram-waveform representation learning method for audio classification task. Three techniques are introduced to solve the challenges in aspect of temporal alignment and semantic alignment problems. Specifically, MSAE model is proposed to align the waveform feature to spectrogram patches. Contrastive learning between spectrogram and waveform representations is proposed as a new pretraining objective. Fusion bottleneck token is introduced for better finetuning performance. System comparison and ablation studies are conducted on the proposed method.

**Strengths:**

1. The proposed method achieves higher or comparable performance on audio classification task compared to the existing SSL-based methods.
2. Sufficient ablation studies are conducted to show the effectiveness of the proposed method and the effect of different hyper-parameters.
3. The idea of patchfying 1d waveform representation to align with the 2d spectrogram representation is somehow novel.

**Weaknesses:**

1. Some statements are inaccurate or unclear.

   a) In introduction paragraph 1, the authors try to illustrate the difference between spectrogram and waveform representation by differentiating the tasks based on them. However, many of the audio/speech tasks can build on both spectrogram and waveform representation and both achieve good results. Actually, in areas like audio signal processing and ASR, both spectrogram [1,2] and waveform [3,4] are frequently used.

   b) In introduction paragraph 2, it is quite confusing why car engine sound is more clear in the time-frequency domain, while successive breezes is clear in waveform domain. Need clarification. Moreover, waveform representation can also present time-frequency patterns. Take an example of conv-TasNet [3] in audio signal processing domain, the waveform filters spontaneously converge to a frequency response similar to that of a log-mel filter bank.

2. Some experimental results / settings are confusing.

   a) To sufficiently prove the effectiveness of spectrogram-waveform representation combination, the authors should show the comparison between spectrogram-only, waveform-only, and joint spectrogram-waveform representations while **keeping other factors the same**. However, the waveform-only results come from very old research, where the SSL and audio transformer techniques are not well established. Since waveform includes more information than the spectrogram, maybe using WaPT will result in better performance than SSaPT and comparable performance of PSWaCL. If this is true, spectrogram can be completely substituted by waveform in audio classification task. Here, WaPT denotes "Waveform modeling in PreTraining" and the only difference form SWaPT is to remove the spectrogram input and the corresponding training loss.

   b) The result of "SWaPT is worse than SSaPT" is confusing. Why adding additional representation degrades performance? If the reason is the conflict between spectrogram and waveform representation, why not use different parameters in waveform branch and spectrogram branch? If this simple way can solve the misalignment between spectrogram and waveform feature, the necessity of PSWaCL is quite doubtful.

[1] Yin, Dacheng, et al. "Phasen: A phase-and-harmonics-aware speech enhancement network." Proceedings of the AAAI Conference on Artificial Intelligence. Vol. 34. No. 05. 2020.

[2] Gulati, Anmol, et al. "Conformer: Convolution-augmented transformer for speech recognition." Interspeech 2020.

[3] Luo, Yi, and Nima Mesgarani. "Conv-tasnet: Surpassing ideal time–frequency magnitude masking for speech separation." IEEE/ACM transactions on audio, speech, and language processing 27.8 (2019): 1256-1266.

[4] Chen, Sanyuan, et al. "Wavlm: Large-scale self-supervised pre-training for full stack speech processing." IEEE Journal of Selected Topics in Signal Processing 16.6 (2022): 1505-1518.

**Questions:**

See weaknesses.

---

> ### Author Response · Authors · 2023-11-21
> **Response to reviewer Cffs [Part 1/2]**
>
> Dear Reviewer Cffs,
>
> Thank you so much for taking the time to read our paper and providing very valuable comments and suggestions regarding the contributions and presentations. Please see our point-by-point responses.
>
> > 1. Some statements are inaccurate or unclear.
> 1a) In introduction paragraph 1, the authors try to illustrate the difference between spectrogram and waveform representation by differentiating the tasks based on them. However, many of the audio/speech tasks can build on both spectrogram and waveform representation and both achieve good results. Actually, in areas like audio signal processing and ASR, both spectrogram [1,2] and waveform [3,4] are frequently used.
>
> While acknowledging that many audio / speech solutions are built upon both spectrogram and waveform, our goal is not to differentiate them or select the best of them. Instead, we would like to simultaneously combine them and draw complementary presentations from each other. In addition, while there are a bunch of spectrogram-waveform fusion / joint modeling works, we would like to note that CRAFT is the first work to contrast spectrogram and waveform and learn joint representations from them.
>
> > 1b) In introduction paragraph 2, it is quite confusing why car engine sound is more clear in the time-frequency domain, while successive breezes is clear in waveform domain. Need clarification. Moreover, waveform representation can also present time-frequency patterns. Take an example of conv-TasNet in audio signal processing domain, the waveform filters spontaneously converge to a frequency response similar to that of a log-mel filter bank.
>
> By elaborating car engine sound and successive breezes, we are explaining the different focus of spectrogram and waveform representations, which are more concentrated on time-frequency and time-amplitude features, respectively. It might be easier to locate the explosive car engine sound by detecting frequency changes, and successive breezes might be more accurately reflected on amplitude changes.
>
> We understand that the reviewer's concern is the redundancy of using spectrograms and waveforms at the same time. However, it has been demonstrated in many works that spectrograms and waveforms contain complementary information. By tailored design of joint-learning on both representations, the performances can be improved compared with single-representation learning, for example [1-3].
>
> Thank you for the constructive comment of one possible waveform augmentation. The main contribution of this paper is to consider the encoded **waveform itself as one way of data augmentation against spectrogram**, and contrast them. The design of data augmentation in self-supervised learning is an active research question. We would like to thank reviewer Cffs for pointing out conv-TasNet, which is intriguing for us to design other audio contrastive pairs in the future.
>
>
> > 2. Some experimental results / settings are confusing.
> 2a) To sufficiently prove the effectiveness of spectrogram-waveform representation combination, the authors should show the comparison between spectrogram-only, waveform-only, and joint spectrogram-waveform representations while **keeping other factors the same**. However, the waveform-only results come from very old research, where the SSL and audio transformer techniques are not well established. Since waveform includes more information than the spectrogram, maybe using WaPT will result in better performance than SSaPT and comparable performance of PSWaCL. If this is true, spectrogram can be completely substituted by waveform in audio classification task. Here, WaPT denotes "Waveform modeling in PreTraining" and the only difference form SWaPT is to remove the spectrogram input and the corresponding training loss.
>
> Thank you for the valuable suggestion, i.e. the ablations of modeling on spectrogram-only, waveform-only, and joint spectrogram-waveform representations, while keeping other factors the same. Accordingly, we have added these results to a new **Appendix A.3** in our revision. Note that as shown in Appendix A.3, modeling on spectrogram have clear advantage over modeling on waveform directly, i.e. **4.6% higher mAP** on AudioSet-bal set. After carefully grid-searched hyper-parameters, **WaPT still has a 9.0\% smaller mAP than CRAFT**.

---

> ### Author Response · Authors · 2023-11-21
> **Response to reviewer Cffs [Part 2/2]**
>
> > 2b) The result of "SWaPT is worse than SSaPT" is confusing. Why adding additional representation degrades performance? If the reason is the conflict between spectrogram and waveform representation, why not use different parameters in waveform branch and spectrogram branch? If this simple way can solve the misalignment between spectrogram and waveform feature, the necessity of PSWaCL is quite doubtful.
>
> Thank you for the valuable suggestion. The question is if we use different parameters in waveform branch and spectrogram branch, i.e. they have 2 individual backbones, will SWaPT be better than SSAST? To answer it, we did a quick experiment on AudioSet-bal pretraining. Let's use SWa2-AST to denote 2 backbone joint-modeling on spectrogram and waveform. The results are given in the following table:
>
>
> | SSAST    | SWa2-AST | SWaPT-AST |  CRAFT |
> | -------- | -------- | -------- | -------- |
> | 13.68    | 14.656     | 14.80     | 14.664 |
>
>
> In this experiment, pretrained on a small dataset, both SWa2-AST and SWaPT-AST successfully perform better than SSAST. However, there is **no significant difference** between them. In particular, sharing same backbone even has higher mAP than having 2 individual backbones. In addition, **2 individual backbones doubles the model parameters** and needs much more memory consumption. Therefore, we chose to report our results of SWaPT-AST instead of SWa2-AST in the paper.
>
> [1] Li, Xinyu, Venkata Chebiyyam, and Katrin Kirchhoff. "Multi-stream network with temporal attention for environmental sound classification." arXiv preprint arXiv:1901.08608 (2019).
>
> [2] Fedorishin, Dennis, et al. Investigating waveform and spectrogram feature fusion for acoustic scene classification. Technical report. Detection, Classification of Acoustic Scenes, and Events 2021.
>
> [3] Yang, Zixiaofan, and Julia Hirschberg. "Predicting Arousal and Valence from Waveforms and Spectrograms Using Deep Neural Networks." Interspeech. 2018.

---

### Official Review · Reviewer_2yfr · 2023-11-01

**Soundness:** 3 good
**Presentation:** 3 good
**Contribution:** 2 fair
**Rating:** 5
**Confidence:** 3

**Summary:**

The paper describes representation learning method both using spectrogram and waveform. Usually, the model takes spectrogram-based feature as an input to the model, or rarely waveform is solely used. However, since the information we can extract from the spectrogram and waveform can be different, it might be better to use both cases as well. To do that, the author basically used a model presented before (which is joint discriminative and generative masked spectrogram patch modeling), and improved this model by adding several techniques to both deal with spectrogram and waveform. In the end, they made an auxiliary loss term using waveform encoder. This waveform encoder uses multi-scale front-end encoder and  the output of the waveform encoder is compared with spectrogram encoder like they are having a different view relationship in SimCLR loss term. Finally, bottleneck fusion method is used to further boost the performance. The result and ablation study showed that the proposed modules are effective in spectrogram and waveform modeling in environmental sound classification task.

**Strengths:**

The most strong part of the paper lies on the model performance. When we see the results in Table 1, we can find that the proposed method reached the best performing model among self-supervised learning approaches. Also, the paper is easy-to-read and written clearly.

**Weaknesses:**

However, I think the novelty of the paper is quite limited. When we see the results in Table 5 (ablation study), the results is quite obvious. It is well-known that the performance is increased if we apply multi-scale modeling on acoustic model. Also, SimCLR loss and bottleneck fusion methods are also quite known approaches.

**Questions:**

If there is more insights we can get from the model, then it would have more novelty on the paper. For example, since the proposed work contains both spectrogram and waveform based encoder, maybe we can compare the learned characteristics of each encoder (especially the waveform encoder is multi-scaled).

---

> ### Author Response · Authors · 2023-11-21
> **Response to reviewer 2yfr**
>
> Dear Reviewer 2yfr,
>
> Thank you so much for taking the time to read our paper and providing valuable comments and suggestions regarding the novelty and more insights of this work. Please see the following for our point-by-point responses.
>
> > However, I think the novelty of the paper is quite limited. When we see the results in Table 5 (ablation study), the results is quite obvious. It is well-known that the performance is increased if we apply multi-scale modeling on acoustic model. Also, SimCLR loss and bottleneck fusion methods are also quite known approaches.
>
> > If there is more insights we can get from the model, then it would have more novelty on the paper. For example, since the proposed work contains both spectrogram and waveform based encoder, maybe we can compare the learned characteristics of each encoder (especially the waveform encoder is multi-scaled).
>
> We understand the reviewer’s concern about the novelty. Just as the reviewer points out, the effectiveness of multi-scale modeling on acoustic model, SimCLR loss and bottleneck fusion methods have been extensively studied. Therefore, we consider the proposed multi-scale MSAE on waveform in the era of transformer as a minor contribution. In addition, we never claim the SimCLR and bottleneck fusion themselves as our contributions.
>
> The main contribution / novelty of this work is **the novel contrastive learning between different representations (spectrogram and waveform), which are indeed calculated from the same data source**. We are the first to demonstrate that in the era of transformer, different representations from same data sample form the natural pair in contrastive learning, which are orthogonal to many existing learning frameworks. After incorporating other carefully designed techniques (MSAE, SimCLR loss and bottleneck fusion), they successfully enhanced the audio modeling performances. **Treating waveform as a data augmentation against spectrograms**, we followed the SimCLR framework and loss. To unleash the pretrained modeling on both spectrogram and waveform, we applied bottleneck fusion. We believe this is a non-trivial finding and want to convey the message to the audio research community that **contrasting spectrogram and waveform could boost the existing audio modeling methods**.
>
> Just as the reviewer pointed out, the learned characteristics of each encoder can be compared. In CRAFT, the waveform embedding is specifically designed as multi-scaled to mimic the log-Mel spectrogram embedding. Due to the similar representation embedding pipelines, the embedded waveform thus can be treated as data augmentation with respect to spectrogram. In consequence, similar as SimCLR, they form the two branches of our transformer backbone.

---

### Author Response · Authors · 2023-11-21
**General response to all reviewers**

We would like to sincerely thank all reviewers for your valuable time and your insightful comments on our paper. We are particularly grateful for recognizing our work as **best performing** (reviewers 2yfr, Cffs), **written clearly** (reviewer 2yfr), **interesting idea** (reviewers 7CEx, aBV8). We have tried our best to revise the paper and explain our results, and provide one-to-one responses to your comments. In our responses, we will quote each comment and provide our responses in regular font. Whenever applicable, we will refer to the changes made in the paper, highlighted with blue texts.

All table numbers, figure numbers, and equation numbers refer to the revised paper unless otherwise specified.

We sincerely hope that the revisions meet your expectations and standards.

**Major modifications:**
1. Following reviewer 2yfr, Cffs, aBV8 and 7CEx's suggestions, we carefully revised the introduction and Section. 3.2 to discuss more related fusion works and highlighted our unique contributions.
2. Following reviewer Cffs's comments, we added experimental results of WaPT in Appendix A.3 to test the waveform-only modeling performances.
3. Following reviewer Cffs's comments, we tested SWa2-AST, i.e. using two individual transformer backbones on spectrogram and waveform.

---

### Meta-Review · Area_Chair_b7iM · 2023-12-06

**Metareview:**

A contrastive learning framework is proposed that tries to align time-domain and frequency domain representation of audio. The main idea is to use multi-scale audio embedding (MSAE), contrastive learning between spectrogram and waveform representations, and the use of bottleneck tokens to effectively extract information from waveform and spectrogram represtnations. The MSAE representation uses patches from the outputs of multiscale convolutions, similar to how patches are extracted from spectrograms in prior work. The authors claim that spectrogram and waveform domains are misaligned, which affects quality when the features are simply concatenated before classification. The contrastive learning framework then tries to use pairs of spectrogram and waveform patches to address this. Finally, for fine-tuning, the authors introduce bottleneck tokens for better co-use of waveform and spectrogram features.

Results show quality gains on AudioSet and ESC50. While the reviewers commented on the strong results compared to other self-supervised learning methods on the specified tasks, there are also some concerns about baselines not being strong enough or coming from older works. The authors provided additional ablations to show the gains of combining waveforms and spectrograms. They also argue that theirs is an SSL strategy, but the stronger baselines are supervised models. But the authors fine-tune the model on the end-task. So it is unclear why the numbers are not directly comparable to other SOTA methods on AudioSet and ESC50.

There are concerns on overall novelty since most of the components of the proposed model have been proposed in prior work, and the modifications proposed are somewhat minor. Without additional insights that the model provides and evaluations supporting generality of the technique on more audio tasks, it is hard to assess the overall quality of the work. The authors argue that using multiple views of the data (spectrogram and waveform) for contrastive loss is the most novel component. The novelty here, especially for an ICLR audience, is limited.

The reviewers also pointed out that some claims about why waveforms and spectrograms are both useful may be unclear. In general, they both carry very similar information as has been shown in prior studies that use waveform vs. features for ASR and enhancement. But there are different advantages to using both. An interesting question to answer here would be why not use other feature representations as well. A related point that was raised was around the clarity of presentation; the use of multiple acronyms, terms not always properly described, and loose claims makes it a little challenging to follow all the details. While the authors have revised the paper based on feedback, the presentation score is generally low for the work.

Overall, the authors, as they claim in the rebuttal, are presenting a very specific strategy for using waveform and spectrogram for audio pre-training. This is too limited in terms of novelty.

**Justification For Why Not Higher Score:**

As discussed in the meta review, limited novelty and comparisons with stronger baselines are significant enough to justify the current score.

**Justification For Why Not Lower Score:**

N/A

---

### Decision · Program_Chairs · 2024-01-16

Reject